# Magnitude of meconium stained amniotic fluid and associated factors among women who gave birth in North Shoa Zone hospitals, Amhara Region, Ethiopia 2022

Mitiku Tefera[1]*, Dagmawit Birhanu[2], Abebe Mihretie[1], Ewunetu Belete[3], Abrham Demis[4], Leweyehu Alemaw[2]

1 Department of Midwifery School of Nursing and Midwifery, Asrat Woldeyes Health Science Campus, Debre Berhan University, Debre Berhan, Ethiopia, 2 Department of Nursing, School of Nursing and Midwifery, Asrat Woldeyes Health Science Campus, Debre Berhan University, Debre Berhan, Ethiopia, 3 Department of Midwifery, Ebenate Hospital, South Gonder Zone, Amhara, Ethiopia, 4 Department of Midwifery, Deber-Berhan Health Science College, Debre Berhan, Amhara, Ethiopia

* mitikutefera2632@gmail.com

## Abstract

### Background

The presence of meconium-stained amniotic fluid is one of the causes for birth asphyxia. Each year, over five million neonatal deaths occur worldwide because of meconium-stained amniotic fluid and other causes, of which 90% are due to birth asphyxia. The aim of this study was to assess the magnitude of meconium-stained amniotic fluid and associated factors among women who gave birth in North Shoa Zone Hospitals, Amhara Region, Ethiopia, 2022.

### Materials and methods

An institutional-based cross-sectional study was employed. We used 610 women who gave birth at North Shoa Zone Hospitals, Amhara region, Ethiopia. The study was conducted from June 8 to August 8, 2022. Recruitment for the study was made using a multistage sampling procedure. Fifty percent of the study hospitals were randomly selected, and proportional allocation was done. Participants were selected from the sampling frame, labour and delivery register book, using a systematic random sampling approach. The first person was selected at random, while the remaining individuals were selected at every two "K" intervals across all hospitals. An interview-administered structured questionnaire and chart review checklist were used to gather the data that were entered into Epi-Data Version 4.6 and exported to SPSS. Logistics regression was employed, and a p-value <0.05 was considered statistically significant.

### Result

The magnitude of meconium-stained amniotic fluid was 30.3%. Women with a normal hematocrit level were 83% less likely to develop meconium-stained amniotic fluid. Women whose mid-upper arm circumference value was less than 22.9cm (AOR = 1.9; 95% CI:

**Data Availability Statement:** All relevant data including our SPSS data set are within the paper and its Supporting Information files.

**Funding:** The author(s) received no specific funding for this work.

**Competing interests:** The authors have declared that no competing interests exist.

1.18–3.20), obstructed labour (AOR = 3.6; 95% CI: 1.48–8.83), prolonged labour $\geq$ 15 hr (AOR = 7.5; 95% CI: 7.68–13.3), premature rapture of membrane (AOR = 1.7; 95% CI: 3.22–7.40), foetal tachycardia (AOR = 6.2; 95% CI: 2.41–16.3), and Bradycardia (AOR = 3.1; 95% CI: 1.93–5.28) showed a significant association with meconium-stained amniotic fluid.

## Conclusion

The present study revealed that the magnitude of meconium-stained amniotic fluid in North Shoa Zone is nearly one-third. A normal hematocrit level is a preventive factor for meconium-stained amniotic fluid, and a MUAC value <22.9 cm, obstructed and prolonged labour, PROM, bradycardia, and tachycardia are factors associated with meconium-stained amniotic fluid.

## Introduction

The first substance in the developing fetus gut is meconium that initiates the baby's first bowel movement. Most healthy babies expel their first meconium within 24–48 hours of birth. This meconium is a viscous, black, green, and yellow-coloured fluid that can be found or created from the intestinal epithelial cells, the colon, and body fluids [1]. The foetal intestine begins to produce meconium in the tenth week of pregnancy, but from the time the foetal intestine begins to mature until delivery, meconium does not come out. When this meconium has seen, it indicates that the baby is in trouble while in the womb, so we can divide it into three stages: when the liquid has a yellow colour, we can divide it as "1st", brown or green as "2nd", and if there is no amniotic fluid at all, we can divide it as "3rd". Meanwhile, 2nd and 3rd grade meconium during delivery suggests that there may be serious or threatening and/or poor foetal outcomes [1, 2].

On the global level, the occurrence of meconium-stained amniotic fluid (MSAF) in full-term pregnant women ranges from 7 to 22%, and this MSAF affects delivery in 5% to 25% of all laboring women [3, 4]. MSAF can lead to neonatal mortality and morbidity due to aspiration into the foetal lung. This stained fluid aspiration has immediate effects like lack of oxygen in the body, birth asphyxia, and neonatal sepsis; if it is untreated, it may lead to long-term effects on the newborn related to chronic respiratory disease [5]. The presence of MSAF has increased maternal complications such as operative delivery, amniotic fluid embolism, intrapartum chorioamnionitis, puerperal endometriosis, and wound infection [6].

In Ethiopia, the magnitude of MSAF ranges from 12.1%–24.7%, which showed a higher magnitude as compared from the global figure. The frequency of operative delivery, birth asphyxia, neonatal sepsis, and neonatal intensive care unit admissions were high in MSAF births [7–9]. The existence of MSAF is caused by a variety of circumstances including induction of labour, medical conditions, the start and length of labour, preeclampsia, obstructed labour, PROM, NRFHR, and late gestational age [1, 7–9]. Previous studies did not suggest the possible intervention to reduce the burden of MSAF. However, the present study can increase alertness of health practitioners to identify high-risk labouring women and helps to take prompt action with an understanding of the magnitude and associated factors of MSAF. Furthermore, the findings of this study came from different health institutions in terms of sample size, study area, study population, variables, and tools. As well, it can help healthcare and other stakeholders in the area build early implementation plans that are locally suited to handle the challenges of MSAF in our community. The study's findings will provide a framework and a starting point for any large-scale studies, as they will.

## Materials and methods

### Study design, setting, and period

An institutional-based cross-sectional study was conducted from June 8 to August 8, 2022. The research was carried out in several hospitals in the North Shoa Zone, including Deber Berhan Comprehensive Specialized Hospital (2-month delivery service 591), Debre Sina Primary Hospital (141), Shewarobit Primary Hospital (224), Ataye Primary Hospital (230), Molale Primary Hospital (126), Mahalmeda General Hospital (198), Midaweremo Primary Hospital (150), Enate General Hospital (174), Deneba Primary Hospital (155), and Arereti Primary Hospital (165).The North Shoa Zone has a population of 1,837,490, and its capital city is Debre-Berhan, it is 130 km from Addis Ababa and 695 km from Bahr Dar, the capital of the Amhara region.

### Population

**Source population.** The source population for this study was all women who gave birth in North Shoa Zone hospitals in the Amhara region of Ethiopia.

**Study population.** The study participant were selected women who gave birth in North Shoa Zone hospitals in the Amhara region of Ethiopia.

### Eligibility criteria

**Inclusion criteria.** All women who gave birth during data collection time with a gestational age of greater than 28 weeks, in the North Shoa Zone Hospitals Amhara region of Ethiopia were included.

**Exclusion criteria.** All women who gave birth during the data collection period who had breech presentation and congenital malformations in the North Shoa Zone hospitals in the Amhara region of Ethiopia were excluded.

### Sample size determination and sampling techniques

The sample size (n) required for the study was calculated using the formula to estimate a single population proportion from previous study [6] with a design effect (314*2 = 628).

Recruitment for the study was made using a multistage sampling procedure. Five of the ten hospitals were randomly chosen, and the following proportional allocation was done. Participants were selected from the labour and delivery register book, which served as the sampling frame, using a systematic random sampling approach. The first person was chosen at random, while the remaining individuals were chosen at every two "K" intervals across all hospitals (Fig 1).

### Variable

**Dependent variable.**

➢ Meconium stained amniotic fluid (MSAF)

**Independent variable.**

➢ Socio-demographic and behavioral factors

➢ Obstetrical factors

➢ Medical illness factors

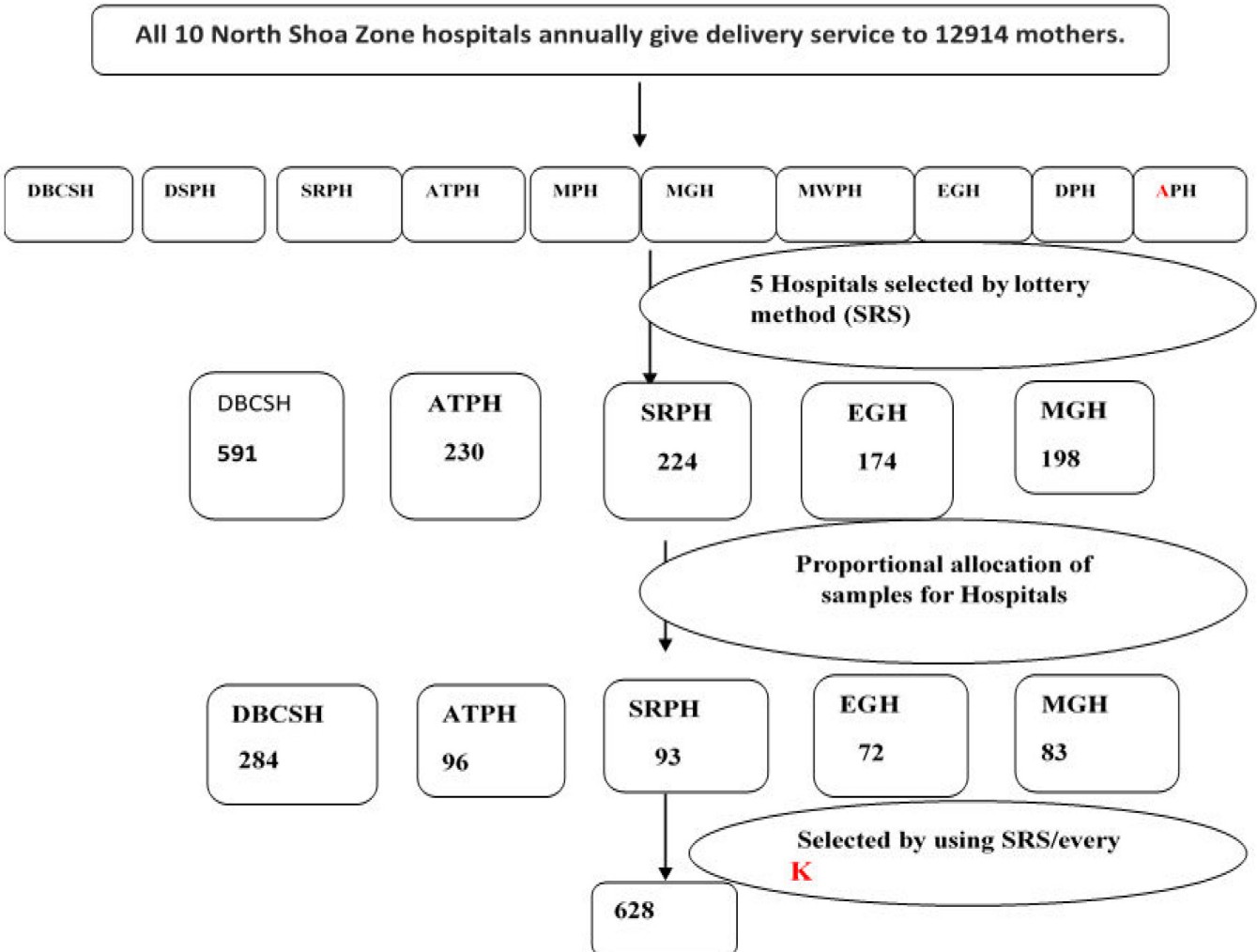

**Fig 1. Schematic presentation of sampling techniques on magnitude of meconium-stained amniotic fluid and associated factors among women who gave birth in North Shoa Zone hospitals, Amhara Region, Ethiopia, 2022.**

## Data collection procedures

The tools for this study were adapted from different previous articles [1, 6, 10, 11]. The data collected using a structured interview-administered questionnaire and chart review. The questionnaire has three sections: behavioural, socio-demographic, obstetrical, and medical characteristics, and so it contains 45 items. Firstly, it was done after a literature review of previously validated and published studies; the English version of the questionnaire was translated to the Amharic version and translated back to the English version to check for its consistency and completeness and to reduce translation errors. The data collector used the Amharic version of the questionnaire for face-to-face interview of women after six hours cesarean section delivery and the remaining questionnaires prepared in English to collect the data from the delivery log-book or chart (show S1 File).

## Data quality assurance and control

Five midwifery professionals trained and employed for the data collection. The questionnaire was distributed to 31 women at Debre-Sinai Hospital (5% of the sample size) before the real data collection began. The main investigator reviewed the data to ensure its completeness and consistency, and we have rectified few sentences on the tool.

## Data processing and analysis

The data was coded and entered in Epi-Data version 4.6 and exported to SPSS. A descriptive analysis was performed on the participant's background characteristics, including behavioral, obstetrical, and medical illnesses. First, bivariable logistic regression analysis was used to select candidate variables for multivariable logistic regression analysis. The variables with a p-value <0.2 in bivariable logistic regression were considered candidate variables and were entered into a multivariable logistic regression model to identify their independent association with the dependent variable. In multivariable analysis, P-values <0.05 were considered statistically significant. The Hosmer-Lemshow goodness test (0.34) and the Nagelkerke R square (0.51) checked the fitness of the regression model, and a multi-collinearity test was done between the independent variables and there is no VIF ($\geq 1$ and $\leq 5$). Finally, results were prepared by using tables, figures, and charts.

## Operational definition

**MSAF:** a confirmed diagnosis written on the chart by the physicians or any other legitimate health professional during the data collection period [5].

**Medical and obstetrical complication:** in this study, the physicians or any other legitimate health professional confirmed all complications.

**Duration of labor:** In this study, a total of the true labour duration of more than 15 hours is considered prolonged labour [6].

**Non-reassuring fetal heart rate pattern:** foetal heart rate of <100 or >180 beats per minute that stays for more than 15 minutes using intermittent auscultations [6].

**Maternal anemia:** If the maternal haemoglobin level is less than 11 g/deciliter [6].

## Ethical consideration

Before we started data collection, we obtained ethical approval from the institutional review board of the Asrat Weldeyes Health Science Campus with protocol number IRB-053. The purpose of the study was explained to the study participants; verbal consent was obtained prior to data collection from all participants; confidentiality was ensured; and at all levels, officials have been contacted and permission has been secured.

## Results

### Socio-demographic and behavioral factors

Data obtained from selected hospitals in the North Shoa Zone for those women who gave birth in the study period, of which 610 responses were obtained, giving us a response rate of 97%. In this study, the median age of the participants was 28 years, with an interquartile range of 7.25. Most of them are married, housewives, and 60% reside in rural areas. From the participants, 24.3% have been drinking alcohol in the last three months, and Khat chewers only in the last three months (Table 1).

**Table 1. Socio-demography characteristics of participants on magnitude of Meconium-stained amniotic fluid and associated factors among women who gave birth in North Shoa Zone, Amhara region, Ethiopia, October 2022 (n = 610).**

| Variables | Category | frequency | Percent (%) |
|---|---|---|---|
| Age of participant | <20 years | 86 | 13.9 |
| | 21–25 years | 154 | 25.3 |
| | 26–30 years | 218 | 35.9 |
| | 31–35 years | 78 | 12.8 |
| | >36 years | 74 | 12.1 |
| Residence | Urban | 244 | 40 |
| | Rural | 366 | 60 |
| Educational status | Cannot write &read | 30 | 4.9 |
| | Non formal education | 262 | 43 |
| | Primary education | 198 | 32.5 |
| | Secondary education | 68 | 11.1 |
| | Tertiary education | 52 | 8.5 |
| Marital status | Married | 519 | 85.1 |
| | Unmarried | 64 | 10.5 |
| | Widowed | 22 | 3.6 |
| | Divorce | 5 | 0.8 |
| Occupation | Students | 51 | 8.4 |
| | House wife | 333 | 54.5 |
| | Governmental employee | 92 | 15.1 |
| | Daily laborer | 6 | 0.9 |
| | Merchant | 108 | 17.7 |
| | NGO | 20 | 3.4 |
| Nutritional statues | MUAC >27.6 | 34 | 5.6 |
| | MUAC 23–27 | 390 | 63.9 |
| | MUAC <22.99 | 186 | 30.5 |
| Chewing within 3 month currently | Yes | 7 | 1.14 |
| | No | 603 | 98.86 |
| Have you drinking Alcohol currently | Yes | 148 | 24.3 |
| | No | 462 | 75.73 |

## Obstetric factors

Among participants, 96.6% had an ANC follow-up, and 94.9% were found to be single-tone pregnant. Of the participants, 80.2% had a term pregnancy, of which 82.4% had a spontaneous onset of labour. Of the participants, 91.1 percent had vertex presentations, and 20 percent were presented with PROM. Besides, 8 percent were anaemic, 94.2% had normal ultrasound findings, and 5% of women received steroid drugs for foetal lung maturity (see Table 2).

## Medical illness

In terms of medical illness, 7.1% of all participants had diabetes, 13% had urinary tract infections, and 4.76% had hepatitis B positive; 90.65% were negative; and 4.59% had unknown hepatitis status (see Table 3).

## Proportion of MSAF

As illustrated in the pie chart below, the magnitude of MSAF was 30.3% (95% CI: 26.9–33.9%). Out of them, 67% had grade one MSAF, 18.5% had grade two, and 14.5% had grade three. The

**Table 2. Frequency of obstetrical characteristics on magnitude of Meconium-stained amniotic fluid and associated factors among women who gave birth in North Shoa Zone, Amhara region, Ethiopia, October 2022 (n = 610).**

| Variables | Category | frequency | Percent (%) |
|---|---|---|---|
| ANC follow-up | Yes | 589 | 96.6 |
| | No | 21 | 3.4 |
| Type of pregnancy | Single | 579 | 94.9 |
| | Multiple | 31 | 5.1 |
| RH statues | Positive | 511 | 83.8 |
| | Negative | 99 | 16.2 |
| Preeclampsia | Yes | 53 | 8.7 |
| | No | 557 | 91.3 |
| Obstructed labour | Yes | 40 | 6.6 |
| | No | 570 | 93.4 |
| APH | Yes | 13 | 2.1 |
| | No | 597 | 97.9 |
| IUGR | Yes | 13 | 2.1 |
| | No | 597 | 97.9 |
| Oligohydraminous | Yes | 28 | 4.6 |
| | No | 582 | 95.4 |
| Poyhydraminous | Yes | 12 | 2.0 |
| | No | 598 | 98.0 |
| GA | Post-term | 12 | 2.0 |
| | Term | 489 | 80.1 |
| | pre-term | 105 | 17.2 |
| | unknown | 4 | 0.7 |
| Onset of labour | Spontaneous | 538 | 88.2 |
| | Induced | 72 | 11.8 |
| Methods of induction (n = 70) | Oxytocin | 72 | 11.8 |
| | Other | 0 | 0 |
| Presentation during labour | Vertex | 556 | 91.1 |
| | Face | 40 | 6.6 |
| | Brow | 14 | 2.2 |
| PROM(before onset of labour) | Yes | 125 | 20.5 |
| | No | 485 | 79.5 |
| Grade of MSAF n = 185 | Grade1 | 124 | 67 |
| | Grade2 | 34 | 18.5 |
| | Grade3 | 27 | 14.5 |
| Stage of L during MSAF n = 185 | Latent phase | 57 | 30.8 |
| | Active phase | 66 | 35.7 |
| | Second stage | 62 | 33.5 |
| Total duration of labour | <14.99 hr | 427 | 70.0 |
| | >15 hr | 183 | 30.0 |
| Mode of delivery | SVD | 503 | 82.45 |
| | C/S | 78 | 12.78 |
| | Instrumental | 29 | 4.77 |
| Fetal heart rate | Tachycardia | 32 | 5.3 |
| | Bradycardia | 160 | 26.2 |
| | Normal | 418 | 68.5 |

*(Continued)*

**Table 2.** (Continued)

| Variables | Category | frequency | Percent (%) |
|---|---|---|---|
| Hematocrit | Normal range | 561 | 91.96 |
| | Anemic | 49 | 8.04 |
| Ultrasonography | Normal | 576 | 94.42 |
| | Abnormal | 27 | 4.44 |
| | Unknown | 7 | 1.14 |
| Received steroid drug (Dexamethasone) | Yes | 31 | 5.09 |
| | No | 579 | 94.91 |

majority of them were diagnosed during the active first stage and the second stage of labour. In all grade two and grade three MSAF cases, 70% were delivered via caesarean section and operative vaginal delivery; the rest of them and grade one MSAF were delivered through spontaneous vaginal delivery (Fig 2).

## Factors associated with meconium-stained amniotic fluid

In this study, the association between socio-demographic, obstetrical, medical, and behavioral factors and MSAF were assessed in all twelve variables to show an association with the dependent variable in the bivariable analysis (show Table 4).

In accordance to the adjusted odd ratio in multivariable logistic regression analysis, the presence of women with lower nutritional status (MUAC) values less than 22.99 cm, the clinical profile of FHR (both tachycardia and bradycardia), hematocrit level, obstructed labour, PROM, and women who had prolonged labour (greater than or equal to 15 hr) were all associated with MSAF. Women with MUAC values of ≤22.99cm were 1.9 (95% CI, 1.18–3.20) times more likely to develop MSAF than women with MUAC values ranging from 23–27.5cm. Similarly, the odds of developing MSAF in women with a NRFHRP, specifically tachycardia, were 6.2 (95% CI, 2.41–16.3) times higher than in women with a normal foetal heart rate pattern, and being Bradycardia was 3.1 (95% CI, 1.9–5.28) times higher than in women with a normal foetal heart rate pattern.

Women who had a normal hematocrit level were 83% less likely to develop MSAF than women who had an anaemic 0.17 (95% CI, 0.07–0.4). Similarly, mothers who experienced

**Table 3. Frequency of medical illnesses factors on magnitude of meconium-stained amniotic fluid and associated factors among women who gave birth in North Shoa Zone, Amhara region, Ethiopia, October 2022 (n = 610).**

| | | | |
|---|---|---|---|
| DM | Yes | 7 | 1.1 |
| | No | 603 | 98.9 |
| Heart disease | Yes | 8 | 1.3 |
| | No | 602 | 98.7 |
| Chronic HTN | Yes | 5 | 0.8 |
| | No | 605 | 99.2 |
| UTI(infection) | Yes | 79 | 13.0 |
| | No | 531 | 87.0 |
| STI(infection) | Yes | 4 | 0.66 |
| | No | 606 | 99.34 |
| HBV status | Positive | 29 | 4.76 |
| | Negative | 553 | 90.65 |
| | Unknown | 28 | 4.59 |

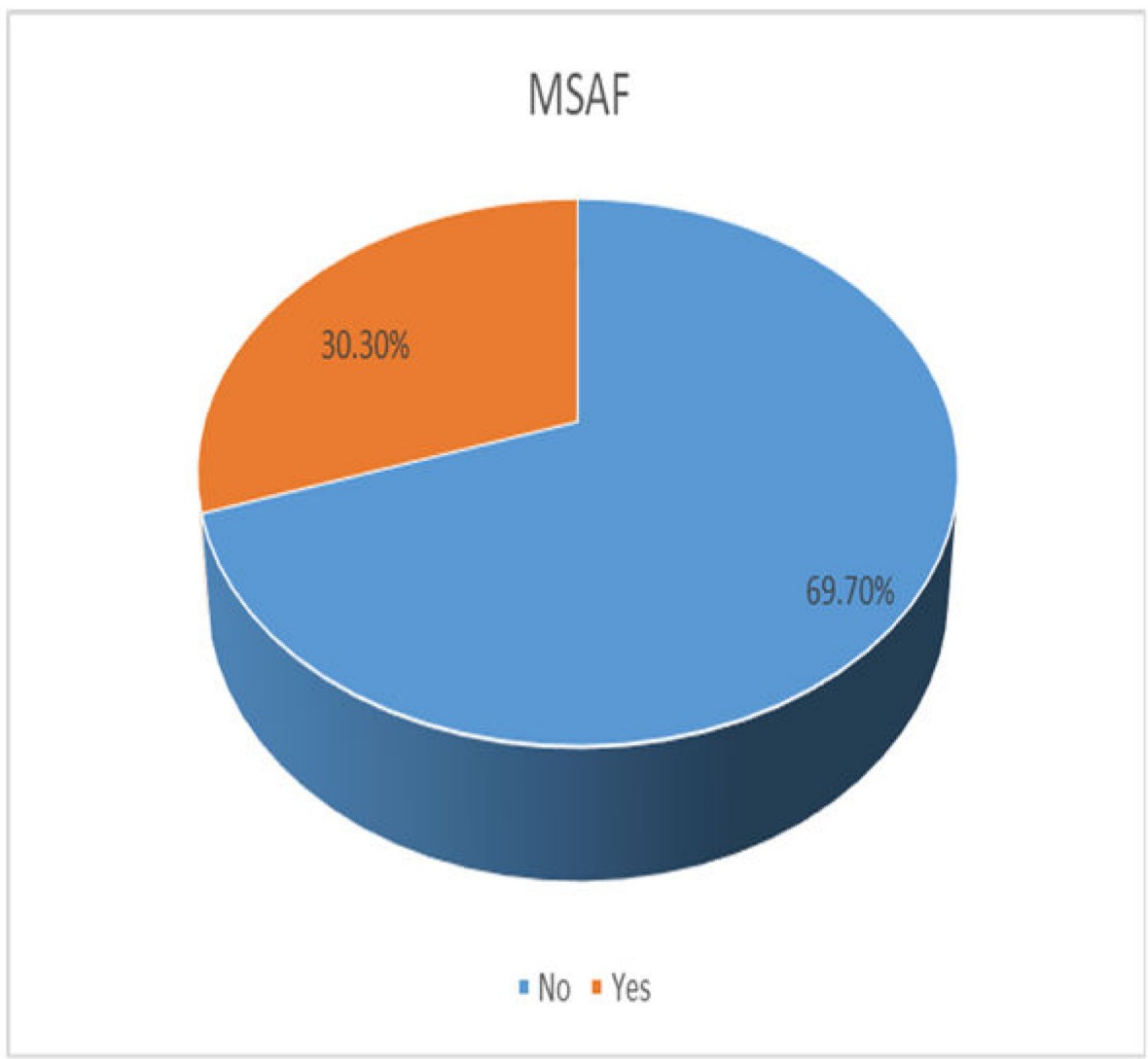

**Fig 2. Proportion of MSAF via pie chart on magnitude of meconium-stained amniotic fluid and associated factors among women who gave birth in North Shoa Zone hospitals, Amhara Region, Ethiopia, 2022.**

prolonged labour ($\geq$15 hours) were 7.5 times more likely to develop MSAF than those who did not (95% CI, 4.64–12.3).

Similarly, the occurrence of MSAF among women who had obstructed labour was 3.6 times more likely to have MSAF compared to women who had no obstructed labour (95% CI, 1.48–8.83). Likewise, the odds of the presence of MSAF in women with PROM were more likely to develop MSAF (1.7; 95% CI, 1.02–3.02) compared to women who did not have premature rapture of the membrane.

## Discussion

The study assessed the magnitude of MSAF and its associated factor in women delivered at North Shoa Zone hospitals and found that 30.3% [95% CI: 26.9–33.9%] had MSAF. The result obtained was higher than a previous study conducted in Bahir-Dar Feleg-Hiwot Hospital (17.8%), another similar study in Bahir-Dar (24.7), Jimma (15.4%), and Addis Ababa (12.1%),

**Table 4. Bi-variable & multivariable logistic regression with COR & AOR on magnitude of meconium stained amniotic fluid and associated factors among women who gave birth in North Shoa Zone, Amhara region, Ethiopia, October 2022(n = 610).**

| Variables | MSAF | | | | | |
|---|---|---|---|---|---|---|
| | Category | Yes | No | COR (95% CI) | AOR(95% CI) | P-value |
| MUAC | > = 27.6 | **10** 1.6% | **24** 3.9% | 1.2(0.53–2.51) | 1.3(0.49–3.66) | 0.568 |
| | < = 22.99 | **72** 12% | **114** 18.6% | 1.7(1.21–2.51) | 1.9(1.18–3.20) | <0.001* |
| | 23–27.5 | **103** 16.8% | **287** 47% | 1 | | |
| ANC follow-up | Yes | **173** 28.3% | **416** 68% | 0.3(0.12–0.75) | 0.5(0.22–2.15) | 0.692 |
| | No | **12** 2.2% | **9** 1.5% | 1 | | |
| Alcohol drinking least3 month | Yes | **66** 10.8% | **82** 13.4% | 2.3(1.57–3.41) | 1.5(0.91–2.65) | 0.106 |
| | No | **119** 19.5% | **343** 56.3% | 1 | | |
| Obstructed Labour | Yes | **27** 4.4% | **13** 2.2% | 5.4(2.72–10.7) | 3.6(1.48–8.83) | 0.005* |
| | No | **158** 26% | **412** 67.5% | 1 | | |
| PROM | Yes | **74** 12.1% | **51** 8.4% | 4.8(3.22–7.40) | 1.7(1.02–3.02) | 0.040* |
| | No | **111** 18.2% | **373** 61.3% | 1 | | |
| Clinical profile of FHR | Tachycardia | **22** 3.7% | **10** 1.6% | 10.9(4.9–24.1) | 6.2(2.41–16.3) | <0.001* |
| | Bradycardia | **93** 15.4% | **67** 10.9% | 6.9(4.6–10.3) | 3.1(1.93–5.28) | <0.001* |
| | Normal | **70** 11.4% | **348** 57% | 1 | | |
| H/C level | Normal | **148** 24.2% | **413** 67.7% | 0.11(0.05–0.22) | 0.17(0.07–0.4) | <0.001* |
| | Anemic | **37** 6.2% | **12** 1.9% | 1 | | |
| Steroid Drug use | Yes | **21** 3.4% | **10** 1.8% | 5.3(2.45–11.52) | 1.8(0.69–5.10) | 0.212 |
| | No | **164** 26.8% | **415** 68% | 1 | | |
| Total duration of labour | ≥15hr | **122** 20% | **61** 10% | 11.5(7.68–13.3) | 7.5(4.64–12.3) | <0.001* |
| | ≤14hr | **63** 10.4% | **364** 59.6% | 1 | | |
| UTI | Yes | **49** 8.2% | **30** 4.9% | 4.7(2.89–7.77) | 1.4(0.75–2.93) | 0.246 |
| | No | **136** 22.2% | **395** 64.7% | 1 | | |
| polyhydramnios | Yes | **8** 1.2% | **5** 0.8% | 3.7(1.22–11.7) | 1.5(0.37–6.77) | 0.526 |
| | No | **177** 29% | **421** 69% | 1 | | |
| oligohydramnios | Yes | **5** 0.8% | **23** 3.7% | 0.5(0.18–1.29) | 0.3(0.12–1.24) | 0.113 |
| | No | **180** 29.6% | **402** 65.9% | 1 | | |

*indicates the variables were significant at P<0.05, **1** = reference group.

Cameron (11.15%), Ghana (8.3%), Brazil (11.9%), India (15.7%), and Israel (5.8%), respectively [1, 6, 8–10, 12–15].

The high result obtained in this investigation may be because of a difference in the study period; the study conducted by Amenu et al 2016 six years ago is not considered as a latest evidence. The lifestyle of the study population was another difference in that the community in North Shoa zone might be different than Bahr Dar and Jimma. This study used several hospitals in the northern Shoa zone, but the other studies were studied at single institutions, which may be one of the main reasons for the high magnitude of MSAF in this study. While the previous studies used only document review to collect data, the current study used interviews and document review to improve the quality of the data. A relatively sufficient sample was used compared to the studies done in Jimma and Addis Ababa. Another difference is that the health systems in Bahr Dar, Jimma, and Addis Ababa are more diverse than the rural areas of the Amhara region in the North Shoa zone. The study in Ghana used only one institution and retrospective data collection from records. Also, the population of the study in Cameron was only taken from pregnant women in one hospital, so these findings were more compressive than previous studies in Ethiopia and Africa. Therefore, it was confirmed that this study showed greater magnitude than studies done in other areas of Ethiopia for all the reasons mentioned above.

In this study, women's nutritional status (MUAC) with a value ≤ 22.99 cm was linked to the presence of MSAF in labouring women. Labouring women with MUAC values of ≤22.99 cm were 1.9 times more likely to develop MSAF than those with MUAC values of 23–27.5 cm. This poor nutritional status is related to poor socioeconomic status, which is inclined to provide inadequate nutrients and oxygen to the unborn foetus and/or can lead to low birth weight and prolonged labour due to insufficient energy to deliver the unborn baby. Because of these reasons, it could be considered a risk factor for the existence of MSAF. This rationale was also shared in other related studies in Ethiopia [6–9, 12].

The presence of a NRFHR pattern was another factor associated with the occurrence of MSAF. Those labouring women whose foetal heart rate was tachycardia were 6.2 times more likely to develop MSAF than women with a normal foetal heart rate, and women whose foetal heart rate was bradycardia were 3.1 times more likely to encounter MSAF as compared to women with a normal foetal heart rate. A study conducted in Pakistan supported this study in that NRFHR was significantly associated with MSAF [13]. A similar association was found in studies conducted at Bahir Dar-Feleg-Hiwot Referral Hospital; additionally, studies in India and Bangladesh revealed a significant association [6, 12, 14]. Nevertheless, a study in Jmma and Addiss Abeba, different from this, suggests that it might be due to sample size variability, which means that when the number of samples increases, there is more chance to diagnose the case. The reason for the presence of MSAF is due to the occurrence of uteroplacental insufficiency followed by foetal hypoxics and the presence of MSAF.

In this study, women with hematocrit levels in the normal range were 83% less likely to develop MSAF as compared to women who were anaemic; this finding was congruent with a study done in India and Lithuania [16, 17]. The plausible reason could be that women who were anaemic may have provoked the foetus to hypoxia inside the womb due to MSAF, which then resulted in neonatal asphyxia related to MSAF. Anaemia can have a negative impact on maternal and foetal health throughout the pregnancy, leading to increased morbidity and foetal death. Anaemia-affected mothers frequently experience breathing difficulties, fainting, tiredness, and palpitations. Hereupon, foetal distress occurred, followed by MSAF [18].

The presence of women with obstructed labour made them 3.6 times more likely to have the chance of developing MSAF as compared to mothers who had not obstructed labour. This finding was consistent with the findings of a study conducted at the Bahir-Dar-Feleg-Hiwot referral hospital in Amhara, Ethiopia. Similarly, this finding was consistent with a study conducted at the Nigerian University Teaching Hospital [1, 19]. The existence of MSAF with obstructed labour leads to the passage of meconium in the foetus; this might be due to inconvenient passengers and passageways during labour and delivery. Another rationale might be due to the possibility of maternal dehydration, maternal distress, and shock. This may result in intrauterine foetal hypoxia secondary to poor placental perfusion and the passage of meconium into the amniotic fluid [1]. The difference between this result and the study done at Bahir-Dar Feleg-Hiwot Referral Hospital might be due to the study setup. The study in Bahir-Dar was done at a single hospital, which means the occurrence of the case might be limited, but this study assessed the factors of MSAF from a different hospital [6].

In this study, prolonged labour duration of ≥15 hours or more was found to have a statistically significant relationship with MSAF. Women who had a long duration of labour were 7.5 times more likely to develop MSAF than women who did not have a long duration of abor. This study was in correspondence with the study results in Nigeria, Bahir-Dar-Feleg-Hiwot referral hospital, and St. Paul's Hospital, Millennium Medical College, Addis Abeba, Ethiopia [6, 11, 20]. The possible clarification for this is explained as follows: when labour duration prolongs, it imposes stress on the foetus, and then the intestine starts to relax to allow passage of meconium. Another possible reason might be that an increased level of corticotrophin-

releasing hormone resulted in increased cortisol during labour, which enhanced myometrial contractions, inducing meconium passage [21].

Women who had PROM before the onset of labour were 1.7 times more likely to develop MSAF than women who did not have PROM before the onset of labour. The finding was consistent with the findings of a study conducted at St. Paul's Hospital Millennium Medical College in Addis Abeba, Ethiopia [11]. The reason could be due to certain factors, like an increase in the risk of maternal infection in women with PROM, such as an increasing ascending infection through the vaginal canal followed by foetal distress resulting in the loss of protection of the umbilical cord by rupture of the membrane and then the presence of meconium in the amniotic fluid (6). Therefore, those identified factors were different from previous studies conducted in Ethiopia.

## Conclusion and recommendation

The present study revealed that the magnitude of meconium-stained amniotic fluid in the North Shoa Zone is nearly one-third. In this study, a normal hematocrit level is a preventive factor for MSAF. Also, a MUAC value <22.9 cm, obstructed and prolonged labour, PROM, bradycardia, and tachycardia were factors associated with meconium-stained amniotic fluid. Those future researchers will investigate a prospective cohort study to get better information about meconium-stained amniotic fluid. Finally, we recommend that if there are complications during labour and delivery and if the mother's hematocrit level is below normal, the health professionals should pay more attention to MSAF, and pregnant women should measure their nutritional status and improve their diet when they come for ANC follow-up.

## Limitation

The study shares some the limitations of a retrospective cross-sectional study design, so it does not show a cause-and-effect relationship between the dependent and independent variables, and it has a social desirability bias in the part variables of behavioral factors.

## Supporting information

**S1 Checklist. Clinical study checklist.**
(DOCX)

**S2 Checklist. STROBE checklist.**
(DOC)

**S1 File. IRB letter.**
(PDF)

**S1 Data. SPSS & STATA data set.**
(SAV)

**S2 Data.**
(DTA)

## Acknowledgments

We would like to express our gratitude to Debre Berhan University, North Shoa Zone Hospitals and staff, study participants, and data collectors for their sincere aid in bringing the necessary information.

## Author Contributions

**Conceptualization:** Mitiku Tefera.

**Data curation:** Mitiku Tefera, Dagmawit Birhanu, Abebe Mihretie, Ewunetu Belete, Abrham Demis, Leweyehu Alemaw.

**Formal analysis:** Mitiku Tefera.

**Funding acquisition:** Mitiku Tefera.

**Investigation:** Mitiku Tefera.

**Methodology:** Mitiku Tefera.

**Project administration:** Mitiku Tefera.

**Resources:** Mitiku Tefera.

**Software:** Mitiku Tefera, Dagmawit Birhanu, Abebe Mihretie, Ewunetu Belete, Abrham Demis, Leweyehu Alemaw.

**Supervision:** Dagmawit Birhanu, Abebe Mihretie, Ewunetu Belete, Abrham Demis, Leweyehu Alemaw.

**Validation:** Dagmawit Birhanu, Abebe Mihretie, Ewunetu Belete, Abrham Demis, Leweyehu Alemaw.

**Visualization:** Dagmawit Birhanu, Abebe Mihretie, Ewunetu Belete, Abrham Demis, Leweyehu Alemaw.

**Writing – original draft:** Mitiku Tefera.

**Writing – review & editing:** Dagmawit Birhanu, Abebe Mihretie, Ewunetu Belete, Abrham Demis, Leweyehu Alemaw.

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
