## [Decision Letter · Decision Letter 0]

11 Apr 2023

PONE-D-22-33578Magnitude of Meconium Stained Amniotic Fluid and Associated Factors Among Women Who Gave Birth in North Shoa Zone Hospital’s Amhara Region Ethiopia 2022PLOS ONE

Dear Dr. Afessa,

Thank you for submitting your manuscript to PLOS ONE. After careful consideration, we feel that it has merit but does not fully meet PLOS ONE’s publication criteria as it currently stands. Therefore, we invite you to submit a revised version of the manuscript that addresses the points raised during the review process.

We look forward to receiving your revised manuscript.

Kind regards,

Hiwot Yisak Dawid, MPH

Academic Editor

PLOS ONE

Journal Requirements

We would like to extend my heartfelt thanks to Debre Berhan University for this research grant.

4. Please amend the manuscript submission data (via Edit Submission) to include author Mitiku Tefera.

5. Please ensure that you refer to Figure 1 in your text as, if accepted, production will need this reference to link the reader to the figure.

Additional Editor Comments:

The sampling techniques used is ambiguous, better to schematically present it and make it scientifically sound.

Reviewers' comments:

Reviewer's Responses to Questions

**Comments to the Author**

1. Is the manuscript technically sound, and do the data support the conclusions?

Reviewer #1: No

Reviewer #2: Yes

2. Has the statistical analysis been performed appropriately and rigorously? 

Reviewer #1: No

Reviewer #2: Yes

3. Have the authors made all data underlying the findings in their manuscript fully available?

Reviewer #1: Yes

Reviewer #2: Yes

4. Is the manuscript presented in an intelligible fashion and written in standard English?

Reviewer #1: No

Reviewer #2: No

5. Review Comments to the Author

Reviewer #1: Thanks for give an opportunity to review this paper.

And I have some comments /concern to the author

1. Most of your study results have a

wide range of AOR . Why it is happen, and also what is the implication in conclusion and recommendation in scientific world?

2.What makes different this paper other than the study design? since there are a number of study conducted in ethiopia.

3.What talents or techniques you used to improve the quality of paper during data collection, data enter and analysis.

4. Do you this the problem is solved after recommend this result for the concerning bodies, due to study design and data analysis that you were exercised.

5. What is the implication of the study in the stud area in particular and in world wide in general.

6. How can co relate for the recommendation of "fetal tachycardia(AOR=6.2; 95% CI; 2.41-16.3) and

Bradycardia (AOR=3.1; 95% CI;1.93-5.28) were significant association with meconium

stained amniotic fluid".

7.how can interpret and possible to recommend by cross sectional study of the following statements:"Women had mid-upper arm circumference value was less than

22.9cm(AOR=1.9; 95% CI;1.18-3.20)".

Thanks

Reviewer #2: Comments to Authors

Abstract

1. Background: Line number 22 &23: Says “In Ethiopia meconium-stained amniotic fluid is under investigated specifically in North Shoa Zone Amhara region Ethiopia”. Do you think that researches for each problem should be done in each zone of the region to say “well investigated”? Why the research findings in other zones of the region like North Gondar, East Gojjam South Wollo…cannot be applicable for North Shoa?

2. Methods: Line 27 &28 : “Two stage cluster sampling was used to recruit study participants”. Is cluster sampling technique appropriate for your research? Why you prefer it over systematic random sampling technique? Would you mind attaching the schematic presentation of the sampling procedure? Why you use a design effect of 2?

3. Conclusion: Line 38: “The magnitude of meconium-stained amniotic fluid, which was high”. What is your ground/reference frame to say “high”? What is “low” to you? Your recommendation is not in line with your finding: Did your research finding showed a gap on “intrapartum service” and “a delay on detecting meconium-stained amniotic fluid “ in the study area?

4. Introduction: line number 45-53: More than 9 lines/ two paragraphs taken from one source without paraphrasing it which is not appropriate writing.

5. Methods and materials: line 80: Better to incorporate how many Hospitals are there in your study setting and number of deliveries in each Hospital monthly.

6. Line 99: I am not comfortable with design effect and cluster sampling technique. I need a strong justification with schematic presentation of sampling procedure.

7. Line 125: what were the adjustments and modifications made to the tool after your pre-test?

8. Lines 137-145: All your operational definitions citation

9. Result: Table 2: Have you ever drinking Alcohol? Yes (69), No (541) and Have you drinking Alcohol currently Yes ?(148), No ( 457). A paradox result,

10. Can we measure behavioural factors with simple yes/no questionnaire?

11. Table: Can it add a value to your result using “gravidity and parity” at the same time as different factors? Look at your result again “Gravidity 2-4=305 and parity 2-4=358, how it could be? Is parity greater than gravidity?

12. Discussion: The literatures compared to your result should be written from local to global unlike that of literature writing style.

6. PLOS authors have the option to publish the peer review history of their article (what does this mean?). If published, this will include your full peer review and any attached files.

Reviewer #1: No

Reviewer #2: **Yes: **Misganaw Fikrie Melesse

---

## [Author Response · Author response to Decision Letter 0]

3 May 2023

Greeting, reviewers We've done all we can to respond to all of your comments, suggestions for improvement, and inquiries. The letter is attached.

---

## [Decision Letter · Decision Letter 1]

2 Oct 2023

PONE-D-22-33578R1Magnitude of Meconium Stained Amniotic Fluid and Associated Factors Among Women Who Gave Birth in North Shoa Zone Hospitals, Amhara Region, Ethiopia 2022.PLOS ONE

Dear Dr. Mitiku Tefera,

Thank you for submitting your manuscript to PLOS ONE. After careful consideration, we feel that it has merit but does not fully meet PLOS ONE’s publication criteria as it currently stands. Therefore, we invite you to submit a revised version of the manuscript that addresses the points raised during the review process.

Please submit your revised manuscript by Nov 16 2023 11:59PM. If you will need more time than this to complete your revisions, please reply to this message or contact the journal office at plosone@plos.org. Please include the following items when submitting your revised manuscript:A rebuttal letter that responds to each point raised by the academic editor and reviewer(s). You should upload this letter as a separate file labeled 'Response to Reviewers'.A marked-up copy of your manuscript that highlights changes made to the original version. You should upload this as a separate file labeled 'Revised Manuscript with Track Changes'.An unmarked version of your revised paper without tracked changes. You should upload this as a separate file labeled 'Manuscript'.

We look forward to receiving your revised manuscript.

Kind regards,

Tamirat Getachew

Academic Editor

PLOS ONE

Reviewers' comments:

Reviewer's Responses to Questions

**Comments to the Author**

1. If the authors have adequately addressed your comments raised in a previous round of review and you feel that this manuscript is now acceptable for publication, you may indicate that here to bypass the “Comments to the Author” section, enter your conflict of interest statement in the “Confidential to Editor” section, and submit your "Accept" recommendation.

Reviewer #3: (No Response)

Reviewer #4: (No Response)

2. Is the manuscript technically sound, and do the data support the conclusions?

Reviewer #3: Yes

Reviewer #4: Yes

3. Has the statistical analysis been performed appropriately and rigorously? 

Reviewer #3: Yes

Reviewer #4: Yes

4. Have the authors made all data underlying the findings in their manuscript fully available?

Reviewer #3: Yes

Reviewer #4: Yes

5. Is the manuscript presented in an intelligible fashion and written in standard English?

Reviewer #3: Yes

Reviewer #4: No

6. Review Comments to the Author

Reviewer #3: Dear respected Editor,

I am thrilled to have the opportunity to review the manuscript entitled “Magnitude of Meconium Stained Amniotic Fluid and Associated Factors among Women Who Gave Birth in North Shoa Zone Hospitals, Amhara Region, Ethiopia 2022”. I have great appreciation for the authors’ efforts to assess the Magnitude of Meconium Stained Amniotic Fluid and Associated Factors Among Women Who Gave Birth in North Shoa Zone Hospitals which may contribute a lot to improve neonatal health. However, I have some concerns which stated as below:

1. Under the Abstract section I have observed a statement which says over five million neonatal deaths occur worldwide due to meconium-stained amniotic fluid. Here may concern is, is these death directly linked to meconium-stained amniotic fluid or as a result of complications associated with MSAF. Since MSAF is no the direct cause of mortality

2. Under the abstract section: there are two ideas; you said you have used two stage of cluster sampling and at the same time also you said you applied systematic random sampling technique to reqruite the study participants. Why did you apply cluster sampling or in your case cluster sampling in not relevant?

3. Under abstract line 29; Fifty percent of the study hospitals were randomly chosen, and following that proportional 30 allocation were done. This statement is not clear. Modify it. Why 50% is chosen?

4. Obstructed labor and prolonged labor: have you checked their collinearities. Not only in terms of statistics but also in biological perspective. Even though the two terms have different definition they have a potential to be collinear with each other. Since if one women diagnosed with obstructed labor her duration of labor is definitely prolonged. How did you see this one?

5. Why this studies is conducted or what makes this studies differ from those conducted previously? Your statement of the problem is too shallow. Please provide rational or justifications for your studies.

6. In your inclusion criteria; you included all those with gestational age of greater than 28 weeks. But MSAF have different implication based on gestational age. For example in case of post term gestation MSAF is may be normal compared to those with preterm gestation. How did you see this issues? And how did you treated those known factor to cause MSAF like induction of labor, mode of deliveries and etc.

7. Under study population: you said the study population was selected women who gave birth North Shoa zone. Those selected women’s are your study participant not your study population please check it and modify accordingly.

8. Check again to your sampling procedure. In your case I don’t think cluster sampling may fit. Of course you applied two stage sampling that’s ok but why you prefer to say cluster sampling?

9. You can add up together table 2 and Table 4 instead of putting only two variable in one table.

10. In discussion section. Better if you not write the confidence interval under discussion

11. Under thine line 259; you said the presence of a NRFHR pattern was associated with the occurrence of MSAF. Which one is correct (MSAF associated with occurrence of NRFHR or NRFHR is associated with occurrence of MSAF?)

12. Under you limitation: I think you reviewed patient’s chart to assess your outcome variable. So you have address it’s drawback or limitation’s under this section

13. Your conclusion should be based on your objectives and major findings.

Reviewer #4: Dear editor thank you so much for your invitation to review this article. The whole work of this study sounds scientific. However, in my point of view, this article doesn’t merit publication in its present status. So I suggest the authors to amend it before online publication. Please, find my comments and suggestions hereunder.

Title: Authors would study an obstetric condition that causes meconium-stained amniotic fluid because it is a sign of underlying obstetric conditions. Fetal distress due to obstructed labour and prolonged labour or other labour complications are the causes of MSAF.

Abstract: line 32 & 33 incomplete sentence.

Introduction

This section would mention the previous interventions undertaken (if any) to lessen the problem or any directed plan against it.

“Based on this study, health practitioners can identify high-risk labouring women and newborns early and take 75 prompt action to improve their outcomes with an understanding of the magnitude and associated factors of 76 MSAF.” Is your study found any MSAF diagnostic modality? What new antecedents of MSAF did your study added to the field? I think all professionals in the field know that anaemia, malnutrition, obstructed and prolonged labour, PROM, bradycardia, and tachycardia predispose to MSAF.

Materials and Methods:

Why did you exclude breech deliveries? Authors missed most important cases because such cases are risky for labour complications which can lead to MSAF. Congenital malformations are also risk factors for labour complications.

The authors mentioned that they used a two-stage cluster sampling. Why cluster sampling? What are the assumptions of cluster? Note that this sampling technique is exceptionally vulnerable for sampling error compared to other probability sampling techniques?

You mentioned that you got the delivery registration book (as a sampling framework) but still you have applied a systematic random sampling technique that less representative than a simple random sampling.

Line 125 to 127 “To reduce translation errors, the data collector used the Amharic version of the questionnaire to collect data from the delivery logbook or chart” Factually, medical and obstetrics histories and reports are documented in English. So how did you collect it in Amharic language and what is its importance? The Amharic language might only be used to interview the mothers.

The data collection techniques where not clearly mentioned. It seems the authors approached the postpartum women. If so, in which interval postpartum period? How did you approached and interviewed very exhausted postpartum mother? You also interviewed women who underwent C/S surgery, so how did you interview those unresponsive women due to the anaesthesia effect? These should be briefed.

What other measurements were taken to maintain data quality? Didn’t the data collectors and supervisors trained?

Authors mentioned that they checked multicolliniarity but they didn’t mention the techniques they used.

How did you measure the degrees of MSAF? Note that it is a subjective observation of the colour and appearances of the AF. So would think the documented diagnosis, observed by different individuals, were reliable?

You didn’t cite sources for your term definitions. Almost all are incorrect citations.

Is it “protocol” or “reference number”?

The ethical clearance and consent procedure was not coherently written.

Line 154 no “Informed consent was obtained for Section 1 (socio-demographic and behavioral factors) from the…” Was your informed consent applied for a certain part of questions? Why?

Authors mentioned that they took a verbal consent from participants. So you would elaborate the procedures you have followed to do that.

Results

There some cells with observed value less than 0.5% in table.

The subsection “Behavioral factors” would be replaced by “Substance use”. And also you would study an important variable ‘smoking’ which is risky for fetal distress and labour complications. Note that smoking causes compromises fetal oxygenation. It also causes hypertrophy due to prolonged placental tissue hypoxia which can lead to placenta previa; in turn leads to labour obstruction. You missed it.

There were missed observations for alcohol drinking. Check Table 2

Some of the variables in table 4 are put in questions forms. Please correct it.

You would include the proportion for each variable the cross tab (table 5).

Discussion

The first elaboration you gave for the higher proportion of MSAF in the area is not plausible. Let you check it.

“Therefore, it was confirmed that this study showed a more truthful picture than studies done in different areas of Ethiopia for all the reasons mentioned above” Please do not use such type expressions.

Most of the authors’ interpretations and elaborations are not plausible. Please, check this sections

As it was a clinical study, authors

Acknowledgment

Authors would acknowledge the study participants, data collectors, and supervisors who played unwavering efforts in the study’s accomplishment.

I think it is not necessary to acknowledge a journal for their peer review service.

Generally, the manuscript is full of grammatical, sentence, and punctuation errors. I recommend the authors correct the mentioned issues. Although the previous reviewers mentioned the same points, you didn’t amend it.

7. PLOS authors have the option to publish the peer review history of their article (what does this mean?). If published, this will include your full peer review and any attached files.

Reviewer #3: **Yes: **Elias Yadeta

Reviewer #4: **Yes: **Wubishet Gezimu

---

## [Author Response · Author response to Decision Letter 1]

13 Oct 2023

We tried to revise it according to your comments, and we are all happy with your suggestions. But we are committed to your next comment.

---

## [Decision Letter · Decision Letter 2]

24 Oct 2023

PONE-D-22-33578R2Magnitude of Meconium Stained Amniotic Fluid and Associated Factors Among Women Who Gave Birth in North Shoa Zone Hospitals, Amhara Region, Ethiopia 2022.PLOS ONE

Dear Dr. Tefera,

Thank you for submitting your manuscript to PLOS ONE. After careful consideration, we feel that it has merit but does not fully meet PLOS ONE’s publication criteria as it currently stands. Therefore, we invite you to submit a revised version of the manuscript that addresses the points raised during the review process.

We look forward to receiving your revised manuscript.

Kind regards,

Tamirat Getachew

Academic Editor

PLOS ONE

**Additional Editor Comments:**

You are expected to address each and every comment or suggestion in detail. Unless you responded to the questions raised by each reviewer and modified your manuscript in a scientifically sound way, it is not accepted for publication. So, rigorously address the given comments and questions accordingly.

Reviewers' comments:

Reviewer's Responses to Questions

**Comments to the Author**

1. If the authors have adequately addressed your comments raised in a previous round of review and you feel that this manuscript is now acceptable for publication, you may indicate that here to bypass the “Comments to the Author” section, enter your conflict of interest statement in the “Confidential to Editor” section, and submit your "Accept" recommendation.

Reviewer #3: All comments have been addressed

Reviewer #4: (No Response)

2. Is the manuscript technically sound, and do the data support the conclusions?

Reviewer #3: Yes

Reviewer #4: (No Response)

3. Has the statistical analysis been performed appropriately and rigorously? 

Reviewer #3: Yes

Reviewer #4: (No Response)

4. Have the authors made all data underlying the findings in their manuscript fully available?

Reviewer #3: Yes

Reviewer #4: (No Response)

5. Is the manuscript presented in an intelligible fashion and written in standard English?

Reviewer #3: Yes

Reviewer #4: (No Response)

6. Review Comments to the Author

Reviewer #3: Thank you for addressing my concern. but Still I have concerns regarding your sampling procedure

You said that the communities living in North Shoa are homogeneous. OK but what about the health facilities found in North shoas zone. Are they homogeneous? Definitely no. because the service and care the women may receive from each facilities not similar because of many factors. What are the assumptions of cluster sampling? If you applied cluster sampling, you have included all participant founded in the selected cluster. But you applied systematic random sampling procedure to recruit the respondents. So it is better if you replace with multistage sampling method.

Reviewer #4: Authors didn't address to all my comments and suggestions point-by-point. In addition, even their responses to some some comments were not scientifically convincing.

7. PLOS authors have the option to publish the peer review history of their article (what does this mean?). If published, this will include your full peer review and any attached files.

Reviewer #3: **Yes: **Elias Yadeta

Reviewer #4: No

---

## [Author Response · Author response to Decision Letter 2]

24 Nov 2023

The reviewer 4 says, 'Authors didn't address all my comments and suggestions point-by-point. In addition, even their responses to some comments were not scientifically convincing'. But, we have revised our manuscript point by point as per his comment. In addition to that, now we amend the comment in depth and are ready for the next comment.

---

## [Decision Letter · Decision Letter 3]

10 Jan 2024

Magnitude of Meconium Stained Amniotic Fluid and Associated Factors Among Women Who Gave Birth in North Shoa Zone Hospitals, Amhara Region, Ethiopia 2022.

PONE-D-22-33578R3

Dear Mitiku Tefera,

We’re pleased to inform you that your manuscript has been judged scientifically suitable for publication and will be formally accepted for publication once it meets all outstanding technical requirements.

Kind regards,

Tamirat Getachew

Academic Editor

PLOS ONE

Additional Editor Comments (optional):

Dear Authors,

I hope you will work on the very minor comments raised by reviewers at the proof stage.

Reviewers' comments:

Reviewer's Responses to Questions

**Comments to the Author**

1. If the authors have adequately addressed your comments raised in a previous round of review and you feel that this manuscript is now acceptable for publication, you may indicate that here to bypass the “Comments to the Author” section, enter your conflict of interest statement in the “Confidential to Editor” section, and submit your "Accept" recommendation.

Reviewer #3: All comments have been addressed

Reviewer #4: (No Response)

2. Is the manuscript technically sound, and do the data support the conclusions?

Reviewer #3: Yes

Reviewer #4: Yes

3. Has the statistical analysis been performed appropriately and rigorously? 

Reviewer #3: Yes

Reviewer #4: Yes

4. Have the authors made all data underlying the findings in their manuscript fully available?

Reviewer #3: Yes

Reviewer #4: Yes

5. Is the manuscript presented in an intelligible fashion and written in standard English?

Reviewer #3: Yes

Reviewer #4: No

6. Review Comments to the Author

Reviewer #3: I satisfied with the Authors responses. I would like to thank all the authors for addressing my comments

Reviewer #4: Authors adequately addressed the majority of my previous comments and suggestions.

However, some of them were not considered by the authors. Like comment on the title. Explanations on the exclusion criteria were not adequate. Please, reconsider these comments.

7. PLOS authors have the option to publish the peer review history of their article (what does this mean?). If published, this will include your full peer review and any attached files.

Reviewer #3: No

Reviewer #4: No

---

## [Editor Report · Acceptance letter]

2 Feb 2024

PONE-D-22-33578R3 

PLOS ONE

Dear Dr. Tefera, 

I'm pleased to inform you that your manuscript has been deemed suitable for publication in PLOS ONE. Congratulations! Your manuscript is now being handed over to our production team.

Kind regards, 

on behalf of

Dr. Tamirat Getachew 

Academic Editor

PLOS ONE